# Can Nutritional Intervention for Obesity and Comorbidities Slow Down Age-Related Hearing Impairment?

**DOI:** 10.3390/nu11071668

**Published:** 2019-07-21

**Authors:** Ting-Hsuan Tang, Juen-Haur Hwang, Ting-Hua Yang, Chuan-Jen Hsu, Chen-Chi Wu, Tien-Chen Liu

**Affiliations:** 1Department of Otolaryngology, National Taiwan University Hospital, Taipei 100, Taiwan; 2Department of Otolaryngology-Head and Neck Surgery, Dalin Tzu Chi Hospital, Buddhist Tzu Chi Medical Foundation, Chiayi 622, Taiwan; 3School of Medicine, Tzu Chi University, Hualien 970, Taiwan; 4Department of Medical Research, China Medical University Hospital, China Medical University, Taichung 404, Taiwan; 5Department of Otolaryngology, Taichung Tzu-Chi Hospital, Taichung 427, Taiwan; 6Department of Otolaryngology, National Taiwan University College of Medicine, Taipei 100, Taiwan

**Keywords:** obesity, age-related hearing impairment, nutrition

## Abstract

Background: Age-related hearing impairment (ARHI), the most common sensory deficit in the elderly, is associated with enormous social and public health burdens. Emerging evidence has suggested that obesity and comorbidities might increase the risk of ARHI. However, no reviews have been published that address the role of nutritional interventions for obesity and comorbidities in the prevention of ARHI. Methods: A PubMed database search was conducted to identify the relationship between obesity and ARHI. “Obesity”, “metabolic syndrome”, “adipose-derived hormone”, “fatty acid”, and “age-related hearing impairment” were included as keywords. Results: A total of 89 articles was analyzed with 39 articles of relevance to ARHI. A high-fat diet may induce oxidative stress, mitochondrial damage, and apoptosis in the inner ear. Statins have been shown to delay the progression of ARHI by improving the lipid profile, reducing oxidative stress, and inhibiting endothelial inflammation. Aldosterone could exert protective effects against ARHI by upregulating the Na-K-2Cl co-transporter 1 in the cochlea. Omega-3 polyunsaturated fatty acids could preserve the cochlear microcirculation by reducing dyslipidemia and inhibiting inflammation. Alpha-lipoic acid and lecithin might delay the progression of ARHI by protecting cochlear mitochondrial DNA from damage due to oxidative stress. Tea and ginseng might protect against ARHI through their anti-obesity and anti-diabetic effects. Conclusions: Nutritional interventions for obesity and comorbidities, including a low-fat diet, supplementation with statins, aldosterone, omega-3 polyunsaturated fatty acids, alpha-lipoic acids, lecithin, tea, and ginseng, may protect against the development of ARHI.

## 1. Introduction

Age-related hearing impairment (ARHI) is the most common sensory deficit in the elderly, worldwide [1]. According to the statistics of the World Health Organization, approximately one-third of the population over 65 years of age suffers from ARHI [2]. The prevalence of ARHI doubles with each decade of life, with 2/3 of the elder adults 70 years old and older with a clinically significant hearing loss, increasing to almost 90% of adults older than 80 years [3]. Recent epidemiologic studies have revealed that ARHI is significantly associated with accelerated cognitive decline [4], dementia [5], depression [6], social isolation [7], hospitalization [8], and increased health care expenditures [9]. Accordingly, ARHI has a tremendous negative impact on the quality of life of affected subjects and constitutes an enormous burden from the social and public health perspectives [10].

Pathologically, ARHI is characterized by progressive deterioration of hearing resulting from degeneration of the cochlea and/or the central auditory pathway [11,12]. ARHI involves an interplay between environmental, medical, genetic, and nutritional factors [12,13,14]. Our previous study in 2009 identified an association between central obesity and ARHI [15]. This observation was subsequently confirmed and validated in several studies that obesity and comorbidities [16], including type II diabetes mellitus (T2DM) [17,18,19,20], cardiovascular disease (CVD) [18,19,20,21], and dyslipidemia [18,19,20,21,22,23], significantly increase the risk for ARHI. However, to the best of our knowledge, no reviews have been published that address the role of nutritional interventions for obesity and comorbidities in the prevention of ARHI. In this article, we reviewed the beneficial effects of nutritional interventions for obesity and comorbidities on physiological auditory function and explored potential preventive and therapeutic options for ARHI.

## 2. Search Strategy

A PubMed database search updated in February 2019 was conducted to identify the relationship between obesity and ARHI. The following keywords were included: “obesity”, “metabolic syndrome”, “adipose-derived hormone”, “fatty acid”, and “age-related hearing impairment”. The search identified a total of 89 articles and 39 articles were of relevance to nutritional intervention for ARHI. Names of the first author, year of publication, sample size, study design, study subjects, and hearing evaluation methods were extracted from the articles. 

## 3. Pathogenetic Mechanisms Linking Obesity and Comorbidities to ARHI

According to established evidence, obesity and comorbidities could contribute to ARHI via three pathogenetic pathways: increased insulin resistance, decreased levels of adiponectin, and dyslipidemia. Figure 1 depicts the potential mechanisms linking central obesity and comorbidities to ARHI.

Obesity and comorbidities contribute to ARHI via increased insulin resistance, decreased adiponectin levels, and dyslipidemia. A high-fat diet might lead to dyslipidemia and induce oxidative stress, mitochondrial damage, and apoptosis in the inner ear. Statins might improve the lipid profile, reduce oxidative stress, and inhibit endothelial inflammation. Aldosterone could upregulate NKCC1 for maintaining the endocochlear potential and high potassium concentration in the endolymph. n-3 PUFA could preserve the cochlear microcirculation by reducing dyslipidemia and inhibiting the inflammatory state. Alpha-lipoic acid and lecithin might protect the cochlear mitochondrial DNA from damage by oxidative stress. Tea and ginseng might protect against ARHI through their anti-obesity and anti-diabetic effects.

### 3.1. Increased Insulin Resistance 

In obese subjects, plasma free fatty acid levels are usually elevated as a consequence of increased release from the hypertrophic adipose tissues and reduced clearance from blood [24]. Increased fatty acid levels are thought to induce insulin resistance, which then results in the development of T2DM [24]. T2DM has been closely related to ARHI, probably owing to neuropathy of the auditory nerve, downregulated Na-K-2Cl cotransporter (NKCC) function, and microangiopathy in the cochlea [25,26]. Morphologically, severe atrophy of the spiral ganglion neurons in the cochlea, myelin degeneration of the eighth nerve, and microangiopathic changes in the endolymphatic sac have been documented in T2DM patients with ARHI [26].

The function of the cochlea depends critically on Na-K-2Cl cotransporter 1 (NKCC1) protein in the epithelial cells of the cochlear lateral wall, particularly the stria vascularis [27]. The lateral wall of the cochlea is responsible for maintaining the endocochlear potential and high K+ concentration in the endolymph [28]. During the aging process, reduced expression of NKCC1 causes a decline in the endocochlear potential [27]. Insulin can upregulate the expression of NKCC1 [29]. Therefore, downregulation of NKCC1 associated with insulin resistance might accelerate the development of ARHI [30,31].

In addition, insulin resistance could reduce endothelial production of nitric oxide, which might inhibit vasodilatation and result in microangiopathy [17,24]. The cochlea is a highly micro-vascular-dependent organ, especially the stria vascularis [17]. Compromised vascular perfusion in the cochlea might contribute to ARHI [32].

### 3.2. Decreased Adiponectin Level

Adiponectin is the most abundant adipokine (i.e., adipose tissue-derived cytokine) in the body [33]. Although adiponectin is primarily produced by adipose tissue, plasma levels of this substance are paradoxically reduced in obese subjects, reflecting obesity-induced adipose dysfunction [34]. Adiponectin has been implicated in several physiological processes and has been reported to exert anti-inflammatory [35], anti-atherogenic [34], anti-diabetic [36,37] and anti-apoptotic [38] effects. In accordance with these physiological functions, decreased plasma adiponectin levels have been found to be associated with cardiovascular disease, kidney disease, hypertension, and metabolic syndrome [39]. In the central nervous system, decreased plasma adiponectin levels have been related to cognitive impairment in humans [39].

In our previous study, we demonstrated a protective role for adiponectin against ARHI in a clinical cohort; wherein higher plasma adiponectin levels were associated with better peripheral hearing function [40]. Further investigations revealed that plasma adiponectin level is modulated by the genotype of adiponectin (*ADIPOQ*). In addition, the interaction between adiponectin and its type 1 receptor in the cochlea might exhibit protective effects against apoptosis in hair cells [41]. These findings are consistent with those of a recent study that demonstrated early-onset hearing loss in adiponectin-knockout mice, which was proposed to result from reduced cochlear blood flow and hair cell apoptosis related to adiponectin deficiency [42].

### 3.3. Dyslipidemia

Central obesity is strongly associated with dyslipidemia, characterized by hypertriglyceridemia and decreased levels of high-density lipoprotein (HDL) cholesterol [43]. Both hypertriglyceridemia and decreased HDL cholesterol have been associated with ARHI [44,45,46]. Dyslipidemia could induce microcirculatory disturbances in the cochlea [47]. Moreover, dyslipidemia might also affect cellular lipid component, contributing to increased production of reactive oxygen species (ROS) [48]. Accumulated oxidative damage in the mitochondria by ROS may then lead to mitochondrial dysfunction and cellular apoptosis [49].

Elevated generation of ROS and subsequent apoptosis has been implicated in various types of hearing loss, including noise-induced, ototoxicity-induced, and ARHI [21,48,49,50]. In a healthy cell, a delicate balance exists between pro- and anti-apoptotic factors, enabling survival and proliferation [51]. In stress situations, such as exposure to noise and ototoxic drugs, and aging, this balance might be disturbed, and specific cells in the cochlea may enter the apoptotic death program. Dyslipidemia associated with obesity might also disrupt the delicate balance and accelerate the progression of hearing loss [44,49,52].

## 4. Nutritional Interventions for Obesity and Comorbidities against ARHI

Parallel to the aforementioned pathogenetic effects of obesity and comorbidities on ARHI, several nutritional interventions have been shown to prevent the development of ARHI, including a low-fat diet, supplementation with statins, aldosterone, omega-3 polyunsaturated fatty acids (n-3 PUFA), alpha-lipoic acids, lecithin, tea, and ginseng (Figure 1). In vitro and in vivo studies were demonstrated in Appendix A. 

### 4.1. Low-Fat Diet

Several studies have suggested that a high-fat diet could increase the risk of ARHI [53,54]. In an animal model using Sprague–Dawley rats, Du et al. showed that a high-fat diet might lead to elevated thresholds of auditory brainstem response (ABR) [55]. Further pathogenetic investigation demonstrated that a high-fat diet induced oxidative stress, mitochondrial damage, and apoptosis in the spiral ganglion and spiral ligament of the inner ear [55].

Vasilyeva et al. investigated the contribution of DM and a high-fat diet to ARHI in middle-aged CBA/CaJ mice [56]. The mice were assigned to three groups: control group, T1DM group, and T2DM group. T1DM was induced by streptozocin injection, and T2DM was induced by feeding the mice with a high-fat diet. In the T1DM group, significant elevation of ABR thresholds and decreased amplitudes of distortion product otoacoustic emissions (DPOAEs) were observed at high frequencies. In the T2DM group, significant elevation of ABR thresholds and decreased amplitudes of DPOAEs were observed at all frequencies. These findings indicated that ARHI was associated with diabetes and a high-fat diet.

Hwang et al. performed similar studies in CD/1 mice, wherein the mice were randomly divided into two groups: a diet-induced obesity group and a control group [57]. Mice in the diet-induced obesity group were fed with a high-fat diet, resulting in higher body weight, fasting plasma triglyceride levels, and omental fat weight. Significant elevation of ABR thresholds at high frequencies was observed in the diet-induced obesity group, and morphological study of the inner ear revealed narrowed blood vessels in the stria vascularis, increased inflammatory changes, and decreased cell density in the spiral ganglion and spiral ligaments. Hwang et al. postulated that a high-fat diet might increase cell death in the inner ear through activating both caspase-dependent and caspase-independent apoptosis signaling pathways. Further pathogenetic studies showed that a high-fat diet might exacerbate hearing degeneration via increased hypoxia, inflammatory responses, and apoptosis of the cochlear cells. 

However, the association between high-fat diet and ARHI could not be recapitulated in C57BL/6J mice, a mouse model that develops sensorineural hearing loss by 12 months of age. Fujita et al. used C57BL/6J mice to explore the effects of a high-fat diet on ARHI [58]. In contrast to previous studies, C57BL/6J mice fed with a high-fat diet did not reveal elevated ABR thresholds and decreased numbers of spiral ganglion cells. The authors attributed the findings to the antioxidant effect of vitamin E in the high-fat diet and the less abnormal glucose tolerance in C57BL6/J mice.

The relationship between high fat intake and ARHI was also investigated in several clinical studies. Gopinath et al. performed retrospective analyses in the 1997–2004 Blue Mountains Hearing Study cohort. Their results confirmed that high dietary intake of cholesterol might increase the risk for ARHI [59]. 

### 4.2. Statins

Statins are the commonly prescribed medication for treating dyslipidemia and preventing cardiovascular diseases [60]. As dyslipidemia has been shown to be associated with ARHI, the protective effects of statins against ARHI have also been explored in several studies. Syka et al. investigated the effects of atorvastatin on ARHI in both C57BL/6J mice and ApoE-deficient mice [61]. As compared to the control groups, atorvastatin-treated mice revealed increased amplitudes of DPOAEs and diminished endothelial inflammatory process, as evidenced by significantly decreased expression of intercellular adhesion molecule 1 (ICAM-1) and vascular cell adhesion molecule 1 (VCAM-1) in the aortic wall. However, serum cholesterol levels did not change in C57BL/6J mice and even increased paradoxically in ApoE-deficient mice, indicating that statins might have lipid-independent effects against ARHI [61,62]. 

In addition to the lipid-lowering effect, it has been demonstrated that statins might exert pleiotropic effects, such as enhancing the production of endothelial nitric oxide and decreasing oxidative stress [63]. As such, statins could likely delay the deterioration of auditory function by eliminating the atherosclerotic endothelial inflammatory process, improving blood flow, and reducing oxidative stress in the inner ear [64].

A clinical study by Gopinath et al. also suggested statins could reduce the risk of developing ARHI [59]. The authors analyzed a total of 274 self-reporting statin users and showed 48% reduced odds of developing ARHI. 

### 4.3. Aldosterone

Aldosterone is a lipid-derived mineralocorticoid that can upregulate the activity of NKCC1 [27]. In addition, aldosterone might have therapeutic effects for ARHI via reduction of post-translational protein modifications, preservation of hair cells and spiral ganglion cells, and partial blocking of apoptotic and cell death pathways [65]. 

Halonen et al. evaluated the effect of long-term systemic aldosterone treatment on ARHI in CBA/CaJ mice [66]. Their results demonstrated upregulation of NKCC1 in the cochlea of mice fed with aldosterone and showed more stable ABR waveforms with aging, indicating that long-term treatment with aldosterone might prevent ARHI.

The clinical study by Tadros et al. also confirmed the role of aldosterone deficiency in the pathogenesis of ARHI [67]. The study included 47 participants over 58 years of age and retrospectively divided them into two groups: the normal-hearing group and ARHI group. Serum aldosterone level was determined in all subjects. A battery of hearing tests was performed, including audiometry, transient evoked otoacoustic emissions (TEOAEs), and hearing in noise test (HINT). In the ARHI group, serum aldosterone levels were significantly lower than in the normal hearing group. Elevated aldosterone levels were correlated with lower pure-tone hearing thresholds and better HINT scores, suggesting that aldosterone might have protective effects against ARHI.

### 4.4. Omega-3 Polyunsaturated Fatty Acids 

Saturated fatty acids are considered atherogenic, whereas polyunsaturated fatty acids have cardioprotective effects [68]. Omega-3 polyunsaturated fatty acids (n-3 PUFA) is a polyunsaturated fatty acid that can regulate serum cholesterol levels by remodeling HDL [69]. It has been reported that n-3 PUFA could prevent diabetic retinopathy through the preservation of microcirculation in the retina [70]. Retinal microvascular abnormalities would be most strongly associated with hearing loss, at least, in women [71]. 

Several studies have reported that n-3 PUFA might protect against the development of ARHI [72]. Martínez et al. fed C57BL/6J mice with n-3 PUFA supplements for 8 months and observed significantly lower ABR thresholds and higher DPOAEs amplitudes in the mice at 10 months [73]. Further investigations revealed that n-3 PUFA supplements could maintain the balance between anti-and pro-inflammatory cytokines, IGF-1 signaling, and compensate for the increased metabolism of homocysteine in the cochlea due to the aging process [73].

Two clinical studies demonstrated similar observations. Gopinath et al. analyzed the association between n-3 PUFA intake and ARHI in the Blue Mountains Hearing Study cohort [74]. The study determined the dietary intake of n-3 PUFA by summing up the intake of eicosapentaenoic acid (EPA), docosapentaenoic acid (DPA), docosahexaenoic acid (DHA), and alpha-linolenic acid. Their results showed the total dietary intake of n-3 PUFA correlated inversely with development of ARHI. Dullemeijer al. investigated the relationship between plasma very-long chain n-3 PUFA levels (i.e., the summation of plasma EPA, DPA, and DHA levels) and ARHI [75]. The study included 720 subjects and grouped them into four quartiles based on the levels of plasma very-long chain n-3 PUFA. Their results showed that subjects with the highest quartile of plasma very-long chain n-3 PUFA levels showed better hearing ability at low frequencies than those with lowest quartile levels, possibly owing to the improved microcirculation in the apical turn of the cochlea.

### 4.5. Alpha-Lipoic Acid

Alpha-lipoic acid is the precursor of EPA and DHA, which are the predominant n-3 PUFA ingredients in fish oil [68]. Seidman et al. investigated the effects of alpha-lipoic acid on ARHI in Fischer rats [76]. Their results showed that alpha-lipoic acid could repair ROS-induced mitochondrial DNA damage in the cochlea and delay the progression of hearing loss in the animals.

### 4.6. Lecithin

Lecithin, a combination of phospholipids with cholesterol-lowering effects [77], is another compound that could preserve the cochlear mitochondrial function. Seidman et al. demonstrated that Fischer rats fed with lecithin for 6 months showed better hearing and less damage to the mitochondrial DNA in the stria vascularis and auditory nerve [78]. Their findings indicated that lecithin, by acting as an anti-oxidant agent, could prevent mitochondrial damage and maintain the mitochondrial membrane potential of cochlear cells in aged animals. 

### 4.7. Phytochemicals 

According to previous studies, several phytochemicals could protect against ARHI. Tea has been shown to have an anti-oxidant effect and potential benefits on hair cell regeneration [79]. Hwang et al. found that the consumption of oolong tea might slow down the central auditory degeneration in the elderly [80]. A clinical study conducted by Hosoda et al. also showed oolong tea could lower plasma glucose level and improve ARHI [81]. It has been proposed that the insulin-like effect of tea polyphenols might be the possible protective mechanism [82]. 

Ginseng also has protective effects against ARHI. C57BL/6 mice fed with ginseng revealed delayed hearing loss and vestibular dysfunction with aging [83]. A similar observation was recapitulated in a T2DM mouse model of C57BL/KsJ background [84]. The beneficial effects of ginseng on the hearing function have been related to the improved insulin sensitivity and protection efficacy of the auditory nerve [84]. In addition, ginseng has also been demonstrated to have anti-obesity effects via decreasing appetite, reducing food absorption, and affecting gut microbiota [85]. 

## 5. Conclusions

Obesity and comorbidities could contribute to ARHI through increased insulin resistance, decreased adiponectin levels, and dyslipidemia. The elucidation of these pathogenetic mechanisms has enabled the development of novel nutritional intervention strategies to protect hearing. Existing evidence indicates that nutritional interventions for obesity and comorbidities, including a low-fat diet, supplementation with statins, aldosterone, n-3 PUFA, alpha-lipoic acids, lecithin, tea, and ginseng, may provide protective benefits against the development of ARHI.

## Figures and Tables

**Figure 1 nutrients-11-01668-f001:**
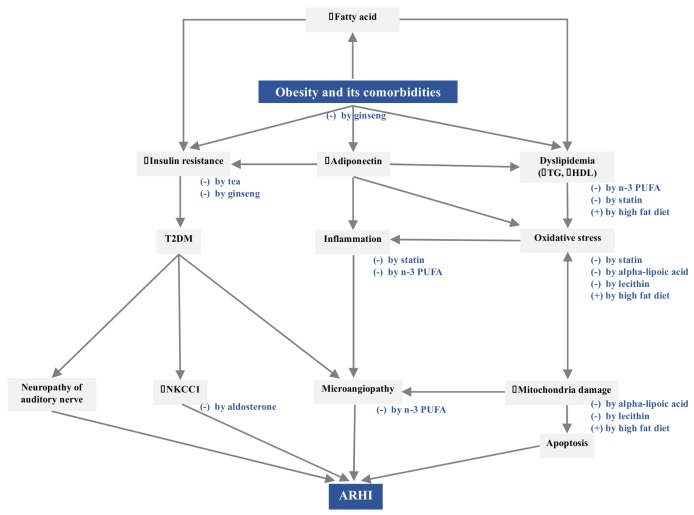
Pathogenetic mechanisms linking obesity and comorbidities to ARHI and the corresponding nutritional interventions. The symbol of (+) indicates stimulation and (-) means inhibition.

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
