# Peer review of "Can Nutritional Intervention for Obesity and Comorbidities Slow Down Age-Related Hearing Impairment?"

_nutrients, 2019, doi:10.3390/nu11071668_

Reviewer 1 Report

In this manuscript, authors review the literature related to pathogenic mechanisms linking obesity and comorbidities to ARHI and nutritional interventions are discussed. Based on the low number of finding published, authors postulate several interventions for obesity and comorbidities as candidates to protect against the development of ARHI. I think this revision has been done quite well but could be further improved.

Minor comments

Figure 1 does not have enough resolution and  legend may be added

Line 48.- This reference may be included

Croll PH, Voortman T, Vernooij MW, Baatenburg de Jong RJ, Lin FR, Rivadeneira F, Ikram MA, Goedegebure A.The association between obesity, diet quality and hearing loss in older adults. Aging (Albany NY). 2019 Jan 4;11(1):48-62.

Search strategy: It would be useful to include the last PubMed revisión date.

Line 77.-I think the reference 26 not match and may be replaced by:

Liu YChu HChen JZhou LChen QYu YWu ZWang SLai YPan CCui Y. Age-related change in the expression of NKCC1 in the cochlear lateral wall of C57BL/6J mice. Acta Otolaryngol. 2014 Oct;134(10):1047-51

Line 85.- reference 29 should  be replaced by these articles

Gates, G. A., Cobb, J. L., D’Agostino R. B., Wolf P. A. (1993). The relation of hearing in the elderly to the presence of cardiovascular disease and cardiovascular risk factors. Archives of Otolaryngology-Head and Neck Surgery, 119, 156–161

Seidman, M. D., Quirk, W. S., Shirwany N. A. (1999). Mechanisms of alterations in the microcirculation of the cochlea. Annals of the New York Academy of Science, 884, 226–232

Line114.- Authors should revise references 48 and 49 because they do not seem to match this ítem.

Line 147.-It should be useful the inclusión of this phrase: Hwang et al postulate that “high-fat diet might increase cell death in the inner ear through activating both caspase-dependent and caspase-independent apoptosis signalling pathways”.

Hwang JH, Hsu CJ, Yu WH, Liu TC, Yang WS. Diet-induced obesity exacerbates auditory degeneration via hypoxia, inflammation, and apoptosis signalling pathways in CD/1 mice. PLoS One. 2013;8(4):e60730. Published 2013 Apr 26.

Line 149.- Authors type “animal models” but they only show data of C57BL/6J, are there any other models not shown?

Líne 174.-Authors state “a recent clinical study” but this study was performed  8 years ago, then recent may be deleted

Line 190.- the following information may be included Frisina et al 2016 proposed a possible cellular and molecular mechanisms for aldosterone therapeutic effect and I think it may be included.

Frisina RD, Ding B, Zhu X, Walton JP. Age-related hearing loss: prevention of threshold declines, cell loss and apoptosis in spiral ganglion neurons.Aging (Albany NY). 2016 Sep 23;8(9):2081-2099..

Line191.-Omega-3 polyunsaturated fatty acids (n-3 PUFA)” is in bold and should be in cursive.

Line 195.- Add retinal microvascular abnormalities would be most strongly associated with hearing loss at least in women.[66]and add this new reference

Shen J, Bi YL, Das UN. Potential role of polyunsaturated fatty acids in diabetic retinopathy.Arch Med Sci. 2014 Dec 22;10(6):1167-74

Reviewer 2 Report

This review intended to address the role of nutritional interventions for obesity and comorbidities in the prevention of age related hearing impairment (ARHI). This manuscript approaches an important and interesting topic that could provide with several ideas on how to design nutritional interventions in obesity and ARHI based on current understanding. However, there are several important aspects of this topic that are not included in this manuscript, such as vitamins, micronutrients (iodine, iron, etc) and anti-obesity and antidiabetic phytochemicals that have shown evidence to delay ARHI (including green tea, panax ginseng, garlic, curcumin, etc).

Specific comments:

Add legend to Figure 1.

In the section “Pathogenetic mechanisms linking obesity and comorbidities to ARHI”, it is advised to provide more information about diabetes-related hearing impairment. Several important reports in animal models and human studies are omitted in this section.

It is advised to compare ARHI and obesity/diabetes pathogenesis separated from obesity/Diabetes-related hearing impairment, in order to provide the reader with a better understanding of the proposed strategy (potential preventive and therapeutic options for ARHI).

Several references are important to be included on the manuscript:

Saito T., Sato K., Saito H. An experimental study of auditory dysfunction associated with hyperlipoproteinemia. Arch Otorhinolaryngol. 1986;243:242–245. doi: 10.1007/BF00464438.

Sikora M.A., Morizono T., Ward W.D., Paparella M.M., Leslie K. Diet-induced hyperlipidemia and auditory dysfunction. Acta Otolaryngol. 1986;102:372–381. doi: 10.3109/00016488609119420.

Curhan S.G., Eavey R.D., Wang M., Rimm E.B., Curhan G.C. Fish and fatty acid consumption and the risk of hearing loss in women. Am. J. Clin. Nutr. 2014;100:1371–1377. doi: 10.3945/ajcn.114.091819.

Author Response

Round  2

Reviewer 2 Report

The authors addressed all the comments properly. I have a few more comments: Authors should add references in this new paragraph: “Obesity and comorbidities contribute to ARHI via increased insulin resistance, decreased adiponectin levels, and dyslipidemia. A high-fat diet might lead to dyslipidemia and induce oxidative stress, mitochondrial damage, and apoptosis in the inner ear. Statins might improve the lipid profile, reduce oxidative stress, and inhibit endothelial inflammation. Aldosterone could upregulate NKCC1 for maintaining the endocochlear potential and high potassium concentration in the endolymph. n-3 PUFA could preserve the cochlear microcirculation by reducing dyslipidemia and inhibiting the inflammatory state. Alpha-lipoic acid and lecithin might protect the cochlear mitochondrial DNA from damage by oxidative stress. Tea and ginseng might protect against ARHI through their anti-obesity and anti-diabetic effects”. Regarding Green tea and ginseng, approached by the authors. Readers might be interested in the evidence of these two supplements in ARHI. Therefore, in the text and in figure 1, the authors could add more information about in vivo and vitro activity and mechanisms, respectively. I suggest the authors check this review, and cite it on the manuscript. Castaneda, R., Natarajan, S., Jeong, S.Y., Hong, B.N., Kang, T.H., 2019. Traditional oriental medicine for sensorineural hearing loss: Can ethnopharmacology contribute to potential drug discovery? J Ethnopharmacol 231, 409-428.
